# Technological Frontiers in Brain Cancer: A Systematic Review and Meta-Analysis of Hyperspectral Imaging in Computer-Aided Diagnosis Systems

**DOI:** 10.3390/diagnostics14171888

**Published:** 2024-08-28

**Authors:** Joseph-Hang Leung, Riya Karmakar, Arvind Mukundan, Wen-Shou Lin, Fathima Anwar, Hsiang-Chen Wang

**Affiliations:** 1Department of Radiology, Ditmanson Medical Foundation Chia-yi Christian Hospital, Chia Yi 60002, Taiwan; 01289@cych.org.tw; 2Department of Mechanical Engineering, National Chung Cheng University, 168, University Rd., Min Hsiung, Chia Yi 62102, Taiwan; karmakarriya345@gmail.com (R.K.); d09420003@ccu.edu.tw (A.M.); 3Neurology Division, Department of Internal Medicine, Kaohsiung Armed Forces General Hospital, 2, Zhongzheng 1st. Rd., Lingya District, Kaohsiung City 80284, Taiwan; 4Faculty of Allied Health Sciences, The University of Lahore, 1-Km Defense Road, Lahore 54590, Punjab, Pakistan; fatimanwarx@gmail.com; 5Department of Medical Research, Dalin Tzu Chi Hospital, Buddhist Tzu Chi Medical Foundation, No. 2, Minsheng Road, Dalin, Chia Yi 62247, Taiwan; 6Department of Technology Development, Hitspectra Intelligent Technology Co., Ltd., 8F.11-1, No. 25, Chenggong 2nd Rd., Qianzhen Dist., Kaohsiung City 80661, Taiwan

**Keywords:** brain cancer, hyperspectral imaging, computer-aided diagnosis, Deek’s funnel plot, forest plot, meta-analysis, diagnostic test accuracy

## Abstract

Brain cancer is a substantial factor in the mortality associated with cancer, presenting difficulties in the timely identification of the disease. The precision of diagnoses is significantly dependent on the proficiency of radiologists and neurologists. Although there is potential for early detection with computer-aided diagnosis (CAD) algorithms, the majority of current research is hindered by its modest sample sizes. This meta-analysis aims to comprehensively assess the diagnostic test accuracy (DTA) of computer-aided design (CAD) models specifically designed for the detection of brain cancer utilizing hyperspectral (HSI) technology. We employ Quadas-2 criteria to choose seven papers and classify the proposed methodologies according to the artificial intelligence method, cancer type, and publication year. In order to evaluate heterogeneity and diagnostic performance, we utilize Deeks’ funnel plot, the forest plot, and accuracy charts. The results of our research suggest that there is no notable variation among the investigations. The CAD techniques that have been examined exhibit a notable level of precision in the automated detection of brain cancer. However, the absence of external validation hinders their potential implementation in real-time clinical settings. This highlights the necessity for additional studies in order to authenticate the CAD models for wider clinical applicability.

## 1. Introduction

According to cancer statistics from 2016 [1], brain cancer is considered the leading cause of mortality around the world. Although significant breakthroughs have been made in the treatment of many other types of cancer [2,3], currently, brain tumors are the leading cause of cancer-related fatalities among children aged 0–14 [4]. The fatality rate due to brain cancer is particularly high in Asia [5,6]. Characterized by uncontrolled tissue growth in the brain, brain tumors can severely impair the body’s normal functioning [5]. There are two main types of brain tumors [7] benign tumors [8,9], which are non-cancerous and generally do not invade nearby tissues or spread, and malignant tumors [10,11], which are cancerous, grow rapidly, and may spread to other parts of the body. Malignant tumors, in turn, can be classified as primary tumors, originating within the brain, or secondary tumors that commonly result from the spread of cancer from other parts of the body [12]. Brain cancer remains difficult to detect, and treating intracranial cancer at its earliest stages [13]. Various types of tumors can produce different symptoms [14]. The proximity of certain brain tumors to critical anatomical structures further complicates diagnosis and treatment [15]. Most often, the first symptom is headache, which can be mild, severe, or come and go. Other symptoms include seizures, loss of balance, nausea, vomiting, disturbed vision or smell, or paralysis in parts of the body.

The diagnosis of brain tumor is the segmentation of tumor regions and the classification of the tumor [16,17], using different methods, which can either be invasive or non-invasive. Biopsy [18,19], the invasive approach, involves collecting a sample of the tumor through incision, and provides crucial information about the tumor’s histological type, classification, and grade [20]. It is not only time consuming and invasive, but also subjective and inconsistent [21]. Modern imaging techniques such as Computed Tomography (CT) [22,23], Magnetic Resonance Imaging (MRI) [24,25], and Positron Emission Tomography (PET) [26,27] are much faster and safer approaches to non-invasive techniques. Despite their efficacy, these imaging scans do not easily allow for the precise quantification of tumor volume due to the accumulation of extracellular water (edema) around the tumor, making accurate discrimination of tumor margins challenging [28].

Based on the statistics from the colorectal cancer center in the United States [29], the incidence of brain cancer and related mortality is consistently rising among individuals aged 4 to 50 [30]. Early diagnosis plays a crucial role in mitigating the severity of brain cancer by facilitating the appropriate treatment. Image processing is a predominant method for identifying brain cancer, with various approaches employing Artificial Neural Networks (ANNs), such as the grey-level co-occurrence matrix (GLCM)-based ANNs, and Residual Neural Network (ResNet) models based on Convolutional Neural Networks (CNNs). Many of the advancements have been made in cancer detection using traditional machine-learning models such as linear discriminant analysis [31], quadratic discriminant analysis [32], support vector machine (SVM) [33,34], decision trees and naïve Bayes [35], k-nearest neighbor’s algorithm [36], k-means [37,38], random forests (RFs) [39], maximum likelihood [40], minimum spanning forest [41], gaussian mixture models [42], semantic texton forest [43], ANNs [44,45], and so on. There are plenty of studies that indicate the superiority of machine-learning algorithms over conventional methods [46,47]. However, challenges arise with these methods due to the substantial dataset requirements for training and issues with handling input transformations. A study by Ewan Gray et al. [48] focused on early economic evaluation for developing a spectroscopic liquid biopsy to detect brain cancer. The research involved 433 blood samples, encompassing cases both with and without brain tumors. Utilizing a fivefold cross-validation strategy for accuracy assessment in the development data, the study reported a sensitivity of 92.8% and a specificity of 91.5%. Yong-Eun Lee Koo et al. investigated several types of nanoparticles for imaging and treating brain cancer [49]. The magnetic nanoparticles based on iron oxide exhibit promising potential as MRI contrast agents in the brain, based on in vitro cellular studies, in vivo animal studies, and human studies. Yan Zhou et al. employed resonance raman (RR) spectroscopy with an excitation wavelength of 532 nm to distinguish between normal brain tissues, cancerous brain tumors, and benign brain lesions. The statistical analysis of RR data initially yielded a diagnostic sensitivity of 90.9% and specificity of 100% [50]. El-Sayed A. El-Dahshan et al. developed a robust classification method for the efficient and automated classification of normal/abnormal brain images [51]. Their algorithm, implemented on a dataset of 101 brain MRI images (14 normal and 87 abnormal), demonstrated an impressive classification accuracy of 99%, sensitivity of 92%, and specificity of 100%.

Despite using the spectroscopic liquid biopsy or developing a computer-aided design (CAD) tool for the detection of brain cancer, these approaches have some major limitations. Hyperspectral imaging (HSI) expands the amount of data gathered beyond what is visible to the human eye, in contrast to traditional RGB (red, green, and blue) imaging. On the other hand, RGB is only limited to recording three diffuse Gaussian spectral bands in the visible spectrum (380–780 nm) [52]. One of HSI’s advantages over other diagnostic technologies is that it is a fully non-invasive, non-contact, non-ionizing, and label-free sensing method [53]. Further experiments and evaluation are therefore desirable to establish whether the proposed approaches have generic applications, while extracting more efficient features and increasing the training dataset. However, the use of HSI is becoming more common around the globe for the diagnosis of cancer. HSI increases the amount of information by capturing more data in many contiguous and narrow spectral bands, over a wide spectral range, and reconstructs the reflectance spectrum for every pixel of the image [54]. HSI has the ability to provide information about different tissue components and their spatial distribution, simultaneously, by measuring the absorption and reflection of light at different wavelengths [55]. Hyperspectral cameras cover different spectral ranges, depending on the type of sensor used. Charge-Coupled Device (CCD) sensors cover the Visible and Near-Infrared (VNIR) range from 400 to 1000 nm, while Indium Gallium Arsenide (InGaAs) sensors can capture HS images in the Near-Infrared (NIR) range, between 900 and 1700 nm [56].

HSI is a technology that combines conventional imaging and spectroscopy to simultaneously obtain the spatial and the spectral information of an object [57]. HSI sensors generate a three-dimensional (3D) data structure, called an HS cube. Spatial information is contained in the first two dimensions, while the third dimension encompasses the spectral information [58]. HS cameras are mostly classified into four different setups, depending on the techniques employed to obtain the HS cube: whiskbroom (point-scanning) cameras, push-broom (line-scanning) cameras, cameras based on spectral scanning (area-scanning or plane-scanning), and snapshot (single shot) cameras [59].

HSI technology has been proved crucial in the field of medicine throughout the years, with various improvements made so that it would effectively contribute to different fields, not only limited to medicine. HS has found comprehensive applications in face recognition [60], remote sensing [61], medical diagnosis [62], archaeology and art conservation [63], vegetation and water resource control, food quality and safety control, forensic medicine [64,65], crime scene detection [66,67], environmental sensing [68], counterfeit hologram [69], glass film detection [70], video capsule endoscopy [71], energy transmission systems, ink mismatch detection in unbalanced clusters [72], lunar penetrating radar [73], security holograms [74], air pollution [75], diabetic retinopathy [76], low-cost holograms [77], counterfeit currency [78], and biomedical areas [79,80]. In order to take advantage of both imaging instruments and offer more helpful information for illness diagnosis and treatment, the HSI system has been combined with many other techniques, such as laparoscope [81], colposcope [82], fundus camera [83,84], and Raman scattering [85]. The most popular combination is with a microscope [86,87], or with a confocal microscope [88], which has shown to be helpful in the investigation of the spectral properties of tissues. Table 1 shows research studies on cancer detection diagnosis using HSI technology in recent years. Figure 1 shows new research on diagnosing brain cancer by combining the advantages and disadvantages of HSI and CAD technology. It presents a thorough summary of the process and essential elements in identifying and diagnosing brain cancer through the use of modern imaging methods and CAD systems. The components symbolize different phases and essential parameters involved in the diagnostic procedure. It highlights the significance of CT and MRI scans as primary imaging modalities for identifying brain disorders. Furthermore, it emphasizes the crucial significance of radiologists and neurologists in analyzing and interpreting these images, underscoring their specialized knowledge and skill in the diagnosis procedure. The presence of a patient receiving an MRI scan exemplifies the practical implementation of various imaging techniques in healthcare environments. In addition, the utilization of AI and HIS technology is necessary for early detection. These sophisticated instruments are illustrated via depictions of neural networks and spectral analysis, demonstrating their function in analyzing intricate data and assisting in the accurate identification of malignant tissues. The workflow demonstrates the interrelated stages, starting with image capture and expert analysis, followed by data manipulation and confirmation of accuracy, highlighting the requirement for external verification and the significance of attaining high sensitivity and specificity in CAD models. Therefore, in this study, a diagnostic meta-analysis has been conducted on CAD studies that diagnose brain cancers using only HSI. A complete QUADAS quality assessment was performed to filter out only quality studies, which were then used to perform Deek’s funnel plot analysis, accuracy analysis, and forest plots.

## 2. Materials and Methods

This section covers the procedures that were involved in obtaining the relevant studies for this review, in particular, studies related to brain cancer detection using HSI technology. The criteria used to choose relevant studies are presented in this section, both for inclusion and exclusion. This review was performed in accordance to the PRISMA (Preferred Reporting Items for Systematic Reviews and Meta-Analyses) guidelines [95]. The overall process of the study is shown in Figure 2. The PRISMA flowchart for the study selection can be seen in Appendix A.

### 2.1. Study Selection Criteria

The purpose of this review is to illuminate the developments in brain cancer diagnosis and detection using HSI, highlighting the system’s strengths and weaknesses in this regard. Studies that satisfy the specified inclusion criteria are the main focus of this review: (1) studies with conclusive numerical results such as dataset, sensitivity, accuracy, precision, and area under the curve (AUC); (2) studies focusing on brain cancer detection using HSI; (3) studies published within the past six years; (4) the publication journal must be in the first quartile and have an H-index greater than 50; (5) studies having prospective and retrospective design; and (6) the studies must be written in English. Furthermore, this review excludes studies that meet the following exclusion criteria: (1) studies with small-scale data investigations; (2) studies that are narrative reviews, systematic reviews, and meta-analyses; (3) comments, proceedings, or study protocols; and (4) conference papers. In this study, two authors introduce the Quality Assessment of Diagnostic Accuracy Studies Version 2 (QUADAS-2) to evaluate the quality of the methodologies in the articles being examined. Bias assessment in patient selection and during index test is included in QUADAS-2 [96]. Additionally, it evaluates the risk bias and the standard of reference in terms of timing and flow as key domains. The applicability assessment was performed in accordance with the PRISMA guidelines, along with the bias assessment. Two authors (R.K. and F.A.) worked independently to perform the QUADAS-2 report, and the results are shown in Figure 3.

**Figure 2 diagnostics-14-01888-f002:**
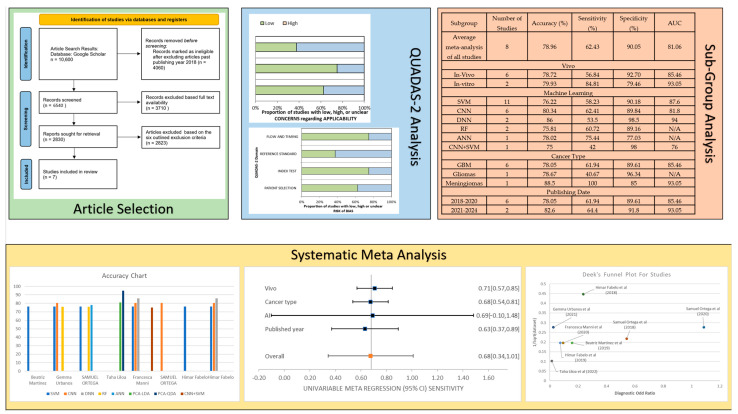
Overall flowchart of the study [32,97,98,99,100,101,102].

### 2.2. QUADAS-2 Results

Table 2 compiles the QUADAS-2 results for the seven studies that are part of this review. It includes the applicability concerns and the degree of bias risk in the studies. Every study was examined for bias risk according to flow and timing, patient selection, reference standard, and index test; applicability issues were examined under patient selection, reference standard, and index test (for the QUADAS-2 domain plot, refer to Appendix A).

## 3. Results

This section presents the findings of the review, along with the brief explanation of each study and the clinical characteristics that were noted. The numerical findings from each study are also included in this section. This section incorporates the comparisons of the results in terms of sensitivity, specificity, and accuracy.

### 3.1. Clinical Features Observed in the Studies

The studies chosen for this article analysis examine the performance of various HSI methods for brain cancer detection and diagnosis. The studies selected for this article analysis briefly outline the performance of various HSI methods for brain cancer detection and diagnosis, as shown in Table 3. Additionally, utilizing subgrouping and meta-analysis, the accuracy, sensitivity, specificity, and area under the curve (AUC) in identifying and categorizing brain cancer lesions and neoplasms from each article were noted. The various HSI techniques employed in the articles were evaluated and contrasted with these indicators.

Martinez et al. used HSI instrumentation to generate the in vivo HSI brain cancer image database. The database employed in this study was composed of 26 HS images from adult patients, and an SVM classifier was used to compare the results. In the 214 and 128 band ranges, the accuracy results stabilized at the value of 80%, with a sensitivity of 55.77% and a specificity of 85.37% [97]. This study included a total of only 26 HS images, which could potentially require external validation before being implemented in a clinical setting. In another study, Urbanos et al. used the SVM, RF, and CNNs across the 660 to 950 nm range of the measured spectrum to classify in vivo brain tissue in thirteen patients with high-grade glioma pathology. The authors found that the SVM achieved 76.5% accuracy, 26% sensitivity, and 91% specificity, on average. On the other hand, RF achieved 82.5% accuracy, 48.5% sensitivity, 99% specificity, while the CNN achieved 82.5% accuracy, 48.5% sensitivity, and 99% specificity. The CNN model had the best result out of the three models used in the study; however, the sensitivity achieved was only around 48.5%, which needs to be significantly increased to be successfully implemented in hospitals. Taking into account the contribution of each band to the global accuracy, the results obtained were 3.81 times higher than the best results of the state-of-the-art models [98].

Ortega et al. processed the data with three different classifiers: SVMs, ANNs, and RF to classify and identify the tissue samples. Twenty-one diagnosed pathological slides obtained from ten different patients were used in this research work. The ANN achieved the best average overall accuracy of 78.02%, the SVM achieved the best average sensitivity of 75.69%, and RF achieved the best average specificity of 79.33%, demonstrating that none of the analyzed classifiers is optimal for all patients [99]. Therefore, the best way to use this method in a hospital would be to combine all three models and to classify with a threshold setting for each of the models. Manni et al. proposed a comparison study using a 2D CNN and two conventional classification methods (the SVM, and the SVM classifier combined with the 3D–2D hybrid CNN for feature extraction). Moreover, the method was compared with the 1D CNN, and a sensitivity of 68%, specificity of 98%, and AUC of 70% for tumor tissue classification were achieved [100]. The study had an impressive specificity of 98%; however, if the sensitivity can be increased, then the study can be implemented in real time. A study conducted by Ortega et al. proposed the use of CNNs for the classification of hematoxylin and eosin (H&E)-stained brain tumor samples. The instrumentation employed in their study consisted of an HS camera coupled to a conventional light microscope, working in the VNIR spectral range from 400 to 1000 nm, with a spectral resolution of 2.8 nm, sampling 826 spectral channels and 1004 spatial pixels. It was concluded that the classification results of HSI provided a more balanced sensitivity of 88% and specificity of 77%, which is the goal for clinical applications, improving the average sensitivity and specificity by 7% and 9% with respect to the RGB imaging results with 81% sensitivity and a specificity of 68% [101].

Fabelo et al. described a methodology using a set of five in vivo brain surface HS images to develop a surgical tool for identifying and delineating the boundaries of the tumor tissue using HS images. The hyperspectral acquisition system employed in their work is called the HELICoiD demonstrator. As a result, the SVM classifier offered specificity and sensitivity results of 99.62% and 99.91%, obtaining overall accuracy results of 99.72%. Even though the overall study achieved impressive results, an external validation is required to actually assess the clinical implantation procedure. In another study, Fabelo et al. employed a VNIR push broom camera to obtain HS images in the spectral range comprised between 400 and 1000 nm. In order to evaluate the deep learning methods against traditional SVM-based machine-learning algorithms, he used the 2D-CNN, 1D-DNN and 1D-CNN methods. The 1D-DNN achieved the best results, obtaining 94% accuracy and 88% sensitivity.

### 3.2. Meta-Analysis of the Studies

Table 4 shows the meta-analysis and subgroup analysis for brain cancer detection. Among the seven studies included in this review, the average obtained accuracy, sensitivity, specificity and AUC were 78.96%, 62.43%, 90.05%, and 81.06%.

While the CNN had the best performance, with 80.34% accuracy, 62.41% sensitivity, 89.84% specificity, and 81.8% AUC, compared to other machine learning, the dataset of six studies showed that the SVM achieved 76.22% accuracy, 58.23% sensitivity, 90.18% specificity, and 87.6% AUC, while the ANN achieved 78.02% accuracy, 75.44% sensitivity, and 77.03% specificity. However, 11 studies used different SVM methods, making the SVM the most used method for the detection and diagnosis of brain cancer according to this evaluation. Himar Fabelo et al. achieved the most promising results using the SVM, with an accuracy of 99.72%, sensitivity of 99.62%, and specificity of 99.91%.

In vivo and in vitro are techniques used by researchers to develop drugs or to study diseases. The term in vivo means research conducted on a living organism, and scientists have been studying the development of machine-learning algorithms using HS images of in vivo brain cancer for the identification of the brain tumor margins. On the other hand, in vitro means research conducted in a laboratory dish or test tube. Each type has benefits and drawbacks. Studies that used the in vitro method showed better results, with an accuracy of 79.93%, sensitivity of 84.81%, specificity of 79.46%, and AUC of 85.46%, as compared to studies that used the in vivo methods, which achieved an accuracy of 78.72%, sensitivity of 56.84%, specificity of 92.70%, and AUC of 85.46%.

High-grade gliomas and glioblastoma are two most prevalent kinds of brain cancer. Gliomas, arising from glial cells in the central nervous system of the adult brain, are the most common primary intracranial tumors and account for 70–80% of all brain tumors. Grade III gliomas and glioblastomas (GBMs) are considered the most aggressive and highly invasive, as they spread quickly to other parts of the brain. Regardless of all the aggressive treatments that include surgery combined with radiation, chemotherapy, and biological therapy, GBM tumors remain a big therapeutic challenge, with survival rates following diagnosis of 12 to 15 months, with less than 3 to 5% of people surviving longer than 5 years. Additionally, GBM tumors have very poor prognosis and are also highly vascular brain tumors. In GBM, there is an upregulation of several angiogenic receptors and factors that trigger angiogenesis-signaling pathways by either downregulating tumor suppressor genes or activating oncogenes. The images of GBM, which are more commonly used by the studies involved in this review, have 78.05% accuracy, 61.94% sensitivity, 89.61% specificity, and 85.46% AUC.

Moreover, an increasing trend was observed in accuracy, sensitivity, and specificity when the studies involved were observed based on their year of publication. This can be illustrated by comparing studies conducted before 2021, which had 78.05% accuracy, 61.94% sensitivity, 61% specificity, and 85.46% AUC, to studies from 2021 to 2023, which had 82.6% accuracy, 64.4% sensitivity, 91.8% specificity, and 93.05% AUC. It was noted that there were fewer studies after 2020, while continuous work was presented in the earlier years. Hence, progressive studies using HSI technology can be beneficial for improving its overall performance in cancer detection and diagnosis.

### 3.3. Subgroup Meta-Analysis

An accuracy graph was generated to visualize the accuracy of different CAD methods in brain cancer detection. Figure 4 shows the overall accuracy chart of different CAD methods used in the studies. The most commonly used method was the SVM, with several types of bands. The use of the SVM in brain cancer detection was the most dominant among the CAD models used in different studies. Consequently, the highest accuracy of 99.72% was achieved by Fabelo et al. through the use of the SVM. By contrast, the lowest accuracy of 53.8 was obtained through the use of band L3 of the SVM, as the CAD method in the investigation by Martinez et al.

Furthermore, Deek’s funnel plot was produced based on different classifications, such as machine-learning methods, vivo, cancer type, and published year in all classifications combined, as shown in Figure 5 (for a detailed Deek’s plot, refer to Appendix A for vivo, AI methods, cancer type, and published year, respectively). Deek’s funnel plot possesses diagnostic odds ratio of each study’s classification and the fraction of the root of each sample size (for sensitivity and specificity computations in the meta-regression, refer to Appendix A). Deek’s funnel plot obtained from this study provided no indication of heterogeneity or bias, with *p* = 0.31 for cancer type. However, Deek’s funnel plot showed heterogeneity, with *p* = 0.003 in all studies and *p* = 0.001 for the CAD methods (for the regression statics of cancer type, vivo, AI methods, publication years and all studies, refer to Appendix A; for the *p*-values for Deek’s funnel plot for seven cancer types, vivo, AI method, publication years and all studies, refer to Appendix A). Finally, forest plots of each CAD method and each study involved for sensitivity and specificity were generated under 95% level of confidence (for the sensitivity and specificity computations in the meta-regression, refer to Appendix A). Top left quadrant in Deek’s funnel plot usually represents studies with small sample sizes and large effect sizes. Studies with large sample sizes and large effect sizes lie in the top right quadrant, while studies with small sample sizes and effect sizes are present in the bottom left quadrant. The bottom right quadrant contains the studies with large sample sizes but small effect sizes. The studies in the top right quadrant suggest the best results; on the contrary, the studies in the bottom left quadrant are believed to be with limited statistical power or inclusive findings. Moreover, a meta-regression analysis was conducted to be able to compare the sensitivities and specificities of each data according to their publication year, cancer type, vivo and AI methods utilized. Figure 6 shows the univariable meta-regression of sensitivity and specificity under a 95% level of confidence interval (for more forest plots see Appendix A for the forest plot of cancer types, AI types, and vivo types, respectively).

## 4. Discussion

The diagnostic accuracy of the studies that were carried out on the detection and diagnosis of brain cancer was found to be “very good” according to Youden’s index, considering the context in which the studies were undertaken. For the purpose of determining the diagnostic accuracy of ill and healthy populations, respectively, sensitivity and specificity at the ideal cut-off point associated with Youden’s index are significant measurements [103]. In accordance with particular DTA (diagnostic test accuracy) requirements, machine-learning algorithms such as the CNN have demonstrated encouraging outcomes in the process of processing medical imaging data for the purpose of cancer detection, including brain cancer [104].

The present clinical recommendations for the management and diagnosis of brain cancer mainly rely on the expertise and interpretation of imaging specialists. This is despite the fact that HSI has shown promising outcomes in the identification of neurological cancer. In the not-too-distant future, it is projected that the application of HSI technology for the identification of brain cancer will achieve broader awareness. In spite of the fact that specialists are hesitant about using them, CAD algorithms have diagnostic skills that are superior to those of standard machine-learning methods.

The guidelines for the identification of brain cancer using neuroimaging techniques have been implemented by the American Society of Neuroradiology (ASNR) [105]. When it comes to the identification of a wide range of brain malignancies, including gliomas and metastatic brain tumors, these criteria require a binding threshold performance that establishes high sensitivity and specificity. In spite of the fact that the overall sensitivity has demonstrated a wide range of values, the specificity requirements for the research that were included in this review were met by only a small number of publications. Even if it is difficult to achieve defined performance standards for brain cancer detection, the incorporation of modern approaches holds promise for improving early diagnosis and treatment outcomes. This is because these techniques have the potential to improve health outcomes.

Several limitations were observed in the research, despite the fact that the diagnostic performance was quite exceptional. The very small number of patients who participated in the studies is one of the limits of these studies. One of the studies that were included in this review had a dataset consisting of five hyperspectral images from five patients. One limitation is a very low dataset of patients involved in the studies. This is a relatively small number when compared to the study that was carried out by Fabelo et al., which had a sample size of thirty-six hyperspectral images from twenty-two different patients. From these data, more than three hundred thousand spectral signatures were associated with the patients. Due to the restricted dataset, there is an increased risk of sampling bias, which has the potential to distort the results and reduce the relevance of the findings. It is possible that this situation will pose a threat to the reliability of the results that were obtained in the study. This is because the dataset might not accurately represent the various features of the population. The fact that only a limited number of cancer types were included is another constraint that contributes to the exacerbation of this problem. This limitation reduces the comprehensiveness of the study. In addition to this, it presents difficulties in applying the findings of the study to a widespread variety of cancer kinds.

In their review analysis, Katharine et al. [106] included around five different categories of cancer research. An additional disadvantage is that there is a limited selection of machine-learning techniques that can be utilized. This may result in the risk of disregarding potentially better procedures that may yield more accurate results. Consequently, this must be avoided. In a comprehensive review of the function of deep learning in brain cancer detection and classification, Nazir et al. [107] highlighted all the machine-learning techniques that were utilized between the years 2015 and 2020. To add insult to injury, the limitation that is imposed by computational power constitutes a significant bottleneck when it comes to undertaking in-depth analysis. The ever-increasing complexity of machine-learning algorithms, in conjunction with the ever-increasing size of datasets, makes it necessary to make use of computational resources in order to facilitate efficient data processing and model training. This could result in an increase in the duration of the experiment, as well as in a reduction in the scalability of the analysis. It is possible to dramatically improve the quality and relevance of future studies in this field by addressing these restrictions through the use of tactics such as expanding the dataset, adding a wider range of cancer kinds, utilizing a variety of machine-learning methodologies, and increasing the amount of computer resources available.

## 5. Conclusions

This systematic review and meta-analysis has provided significant insights into the capabilities and limitations of hyperspectral imaging (HSI) technology in brain cancer detection. We observed that while HSI offers substantial promise in enhancing diagnostic accuracy, challenges remain in terms of limited sample sizes and the generalizability of the study results. To overcome these hurdles, we recommend a concerted effort towards conducting larger, multicentric studies that would enable a more robust statistical analysis and help validate the findings across different demographics and clinical environments. Moreover, integrating HSI with existing diagnostic frameworks could potentially streamline workflows and enhance the decision-making process in clinical settings. This integration should focus on harnessing advanced machine-learning algorithms that can efficiently process the complex data generated by HSI, thereby reducing the burden on clinicians and potentially leading to faster and more accurate diagnosis. In conclusion, while the journey of incorporating HSI into routine clinical practice is still at a nascent stage, our findings underscore its transformative potential in oncological diagnostics. Future research should aim at addressing the current limitations, fostering technological advancements, and ultimately, facilitating the adoption of HSI in clinical practice to improve the outcomes for patients with brain cancer.

## Figures and Tables

**Figure 1 diagnostics-14-01888-f001:**
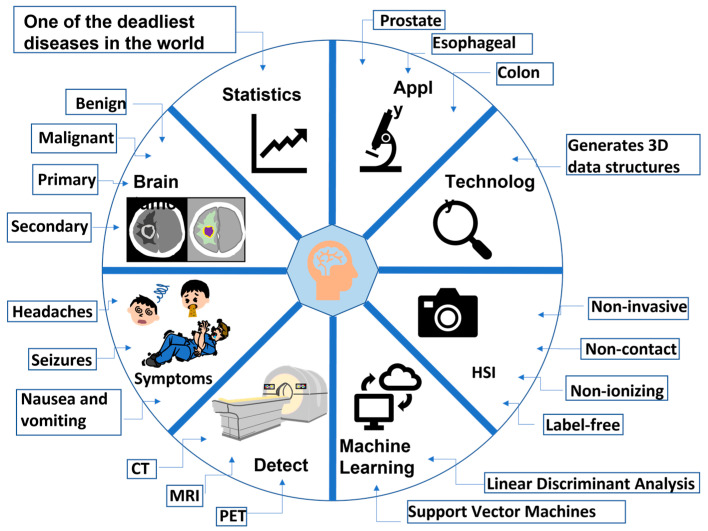
New research on diagnosing brain cancer by combining the advantages and disadvantages of HSI and CAD technology.

**Figure 3 diagnostics-14-01888-f003:**
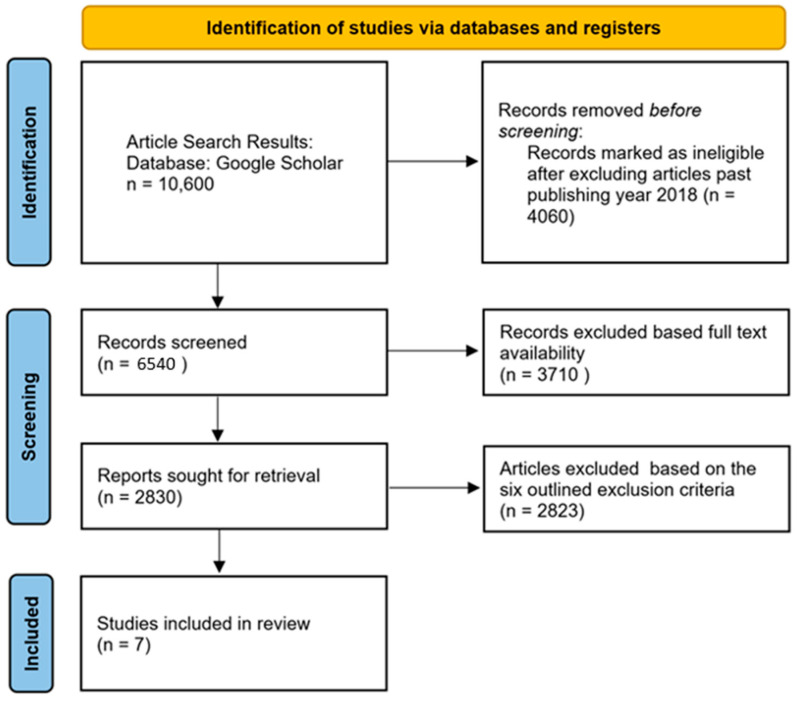
PRISMA 2020 flow diagram.

**Figure 4 diagnostics-14-01888-f004:**
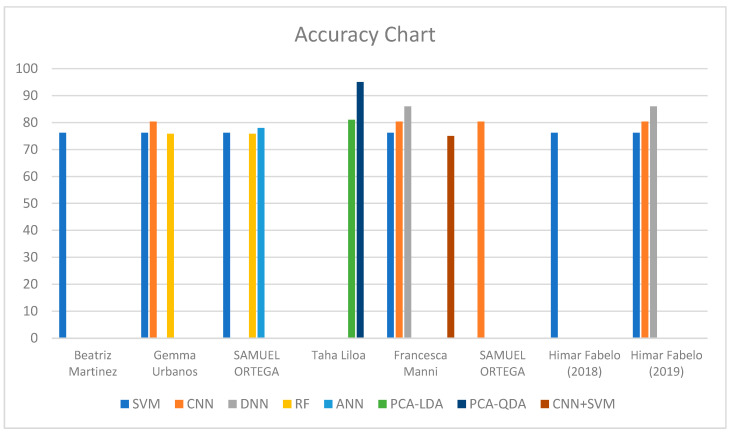
Overall accuracy performance of CAD methods. Different colors represent the methods used in the study [32,97,98,99,100,101,102].

**Figure 5 diagnostics-14-01888-f005:**
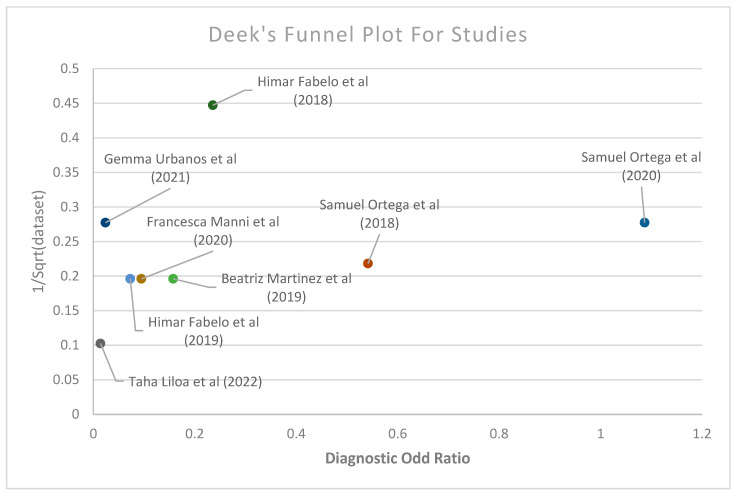
Deeks’ funnel plot of the studies used in the meta-analysis [32,97,98,99,100,101,102].

**Figure 6 diagnostics-14-01888-f006:**
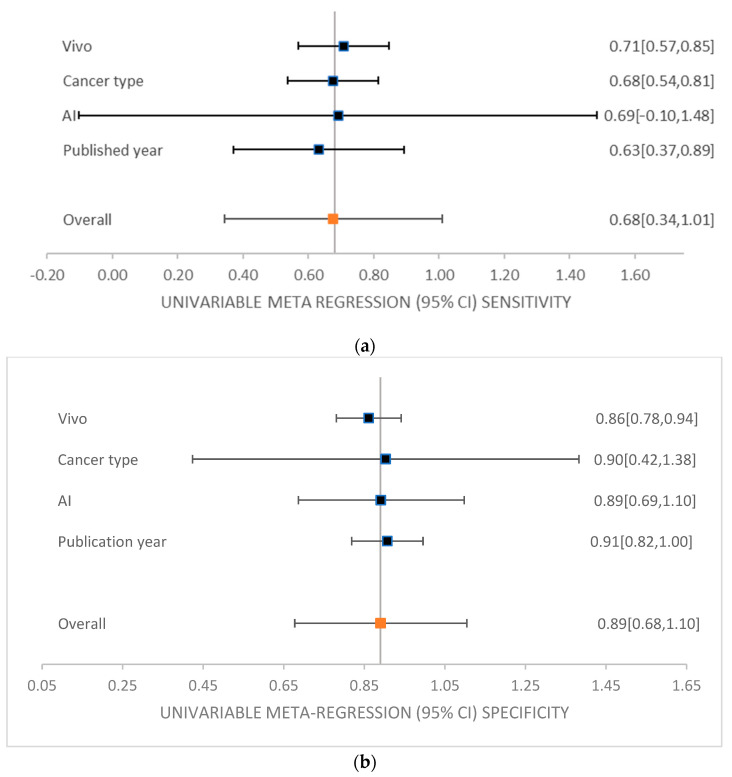
Univariable meta-regression of different subgroup analysis including vivo, cancer type, AI, and publication year. (**a**) Sensitivity, (**b**) specificity.

**Table 1 diagnostics-14-01888-t001:** Studies on HSI in cancer detection.

Year	Author	Application	Spectral Range (nm)
2012 [34]	Hamed Akbari et al.	Prostate cancer	500 to 950 nm
2017 [32]	Guolan Lu et al.	Head and neck cancer	450 to 900 nm
2020 [89]	Francesca Manni et al.	Colon cancer	up to 1700 nm
2022 [90]	Tsung-Jung Tsai at al.	Esophageal cancer	415 and 540 nm
2015 [91]	Atsushi Goto et al.	Gastric cancer	1000 to 2500 nm
2021 [92]	Xuehu Wang et al.	Liver cancer	400–1000 nm
2020 [93]	Ibrahim H et al.	Breast cancer	400–700 nm
2023 [94]	Riheng Chen et al.	Blood cancer	400–1000 nm

**Table 2 diagnostics-14-01888-t002:** QUADAS-2 results of the studies.

Study	Risk of Bias	Applicability Concerns
Patient Selection	Index Test	Reference Standard	Flow and Timing	Patient Selection	Index Test	Reference Standard
Beatriz Martinez et al., 2019 [97]			?		?		?
Gemma Urbanos et al., 2021 [98]			?				?
Samuel Ortega et al., 2018 [99]	?						
Francesca Manni et al., 2020 [100]			?		?		?
Samuel Ortega et al., 2020 [101]	?				?	?	
Himar Fabelo et al., 2018 [32]		?	?			?	?
Himar Fabelo et al., 2019 [102]		?	?	?		?	?


 Low Risk, ? Unclear Risk, 
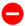
 High Risk.

**Table 3 diagnostics-14-01888-t003:** Clinical features of the studies considered in this review, including the nationality, method, lighting, accuracy, sensitivity, specificity, the number of images, and AUC of the studies.

Author Year	Nationality	Method	Band	Vivo	Index Number	Dataset	Accuracy (%)	Sensitivity (%)	Specificity (%)	AUC (%)
Beatriz Martinez et al., 2019 [97]	Western	SVM band L1	VNIR	In Vivo	1	26	77.9	52.7	94.6	N/A
		SVM band L2			2		77	57	91.2	
		SVM band L3			3		53.8	57.6	70.3	
Gemma Urbanos et al., 2021 [98]	Western	SVM	VNIR	In Vitro	4	13	76.5	26	91	N/A
		RF			5		82.5	48.5	99	
		CNN			6		77	47.5	99	
SAMUEL ORTEGA et al., 2018 [99]	Western	SVM	VNIR	In Vivo	7	21 biopsies	75.53	75.69	70.97	N/A
		ANN			8		78.02	75.44	77.03	
		RF			9		69.13	72.94	79.33	
Francesca Manni et al., 2020 [100]	Western	3D–2D CNN	VNIR	In Vivo	10	26	80	68	98	70
		3D–2D CNN + SVM			11		75	42	98	76
		SVM			12		76	43	98	70
		2DCNN			13		72	14	97	71
		1DDNN			14		78	19	97	89
SAMUEL ORTEGA et al., 2020 [101]	Western	CNN HIS	VNIR	In Vivo	15	13 biopsies	85	88	77	87
		CNN RGB			16		80	81	68	84
Himar Fabelo et al., 2018 [32]	Western	SVM	VNIR	In Vivo	17	5	99.72	99.62	99.91	N/A
Himar Fabelo et al., 2019 [102]	Western	1D-DNN	VNIR	In Vivo	18	26	94	88	100	99
		2D-CNN			19		88	76	100	97
		SVM RBF Opt.			20		84	68	100	97
		SVM RBF Def.			21		73	58	88	86
		SVM Linear Opt.			22		77	54	100	99
		SVM Linear Def			23		68	49	88	86

**Table 4 diagnostics-14-01888-t004:** Subgroup and meta-analysis of diagnostic test accuracy, including the classification of data based on nationality, machine-learning model, images, brain cancer type, and publication date.

Subgroup	Number of Studies	Accuracy (%)	Sensitivity (%)	Specificity (%)	AUC (%)
Average meta-analysis of all studies	8	78.96	62.43	90.05	81.06
Vivo					
In vivo	6	78.72	56.84	92.70	85.46
In vitro	2	79.93	84.81	79.46	93.05
Machine Learning					
SVM	11	76.22	58.23	90.18	87.6
CNN	6	80.34	62.41	89.84	81.8
DNN	2	86	53.5	98.5	94
RF	2	75.81	60.72	89.16	N/A
ANN	1	78.02	75.44	77.03	N/A
CNN + SVM	1	75	42	98	76
Cancer Type					
GBM	6	78.05	61.94	89.61	85.46
Gliomas	1	78.67	40.67	96.34	N/A
Meningiomas	1	88.5	100	85	93.05
Publishing Date					
2018–2020	6	78.05	61.94	89.61	85.46
2021–2024	2	82.6	64.4	91.8	93.05

## Data Availability

The data presented in this study are available in this article upon considerable request to the corresponding author (H.-C.W.). The current study has not been registered; however, upon request to the corresponding author, all the details regarding this systematic review will be provided, along with the template data collection forms, the data extracted from the included studies, the data used for all the analyses, the analytic code, and any other materials used in the review.

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
