# Peer review of "Technological Frontiers in Brain Cancer: A Systematic Review and Meta-Analysis of Hyperspectral Imaging in Computer-Aided Diagnosis Systems"

_diagnostics, 2024, doi:10.3390/diagnostics14171888_

Round 1
Reviewer 1 Report
Comments and Suggestions for Authors
The manuscript provides a thorough analysis of the use of HSI for diagnosing brain cancer. Nevertheless, I have several reservations about the paper.
1. Figure 2 would benefit from a clearer representation, as the current rendition is essentially illegible.
2.The paper could include more technical details about the HSI technology itself. For instance, what is the resolution of the imaging?
3. More discussion of the study’s clinical implications can be beneficial.
4. The discussion on the clinical implications of the findings could be expanded.
5. The manuscript contains an overwhelming amount of citation in the background section, at least over a hundred references. A significant number of these references are likely not relevant to the central topic of the paper.
6. The paper demonstrates a high level of textual similarity.
7. Figure 1 does not convey the message the authors intend to express. Please add a detailed explanation or reconstruct the figure.
8. The references contain quite some formatting issues.
Comments on the Quality of English LanguageN/A
Reviewer 2 Report
Comments and Suggestions for Authors
- This is review paper and any review paper is of great interest to the research community.
- Brain cancer is an important application for the medical community.
- Sometimes you refer in the paper you are doing you review on seven and sometimes you refer you are doing the review on 8 papers . This is not consistent and needs to be rectified.
- stating that this is a review paper snd the same time studying only the results of 7 or 8 methodologies is not enough. For a review you need to include more publications and work.
- Sometimes you display the tables / figures but you don't reference them properly / at all in the text. Your manuscript needs proof reading to correct these mistakes.
- Some figures needs to be re-done as the text in them is not readable.
- You say you are studying only 7 / 8 papers from the abstract , though you reference other papers in the results section. This is not consistent , either you limit your paper to the 7/8 mentioned papers or you increase the number of papers you are referencing all over the sections of your manuscript.
- Sometimes you state some results in the text that are not displayed in the summary table : ex: This meta-analysis suggests that PCA–QDA was the leading technique for brain can- 254 cer detection by terms of accuracy, sensitivity, specificity, and AUC of 95%, 100%, 96%, 255 and 97.5%.. Where are these results in table 3 ?
- In Figure 3 you mention the Himar Fabelo work twice without specifying which work you are referencing to (you need to include the publication year in this case to avoid any confusion).
Comments on the Quality of English LanguageThe paper is written in a clear way but needs read proof to correct some grammatical mistakes / missingness in the content (ex Table without the table number)
Round 2
Reviewer 1 Report
Comments and Suggestions for Authors
Many citations are not related the theme of the manuscript. The need for those citations are absent.
Comments on the Quality of English LanguageN/A
Reviewer 2 Report
Comments and Suggestions for Authors
- Some figures need to be redone as the text in them is not readable: For this comment honestly I do not see any changes between the old figures (exp Fig 2) and the one you state you rectified. The text is still blurry and unclear to me . I see some exact same figures (exp Fig 2) you had in version 1 of your manuscript.
- In Figure 3 you mention the Himar Fabelo work twice without specifying which work you are referencing (you need to include the publication year in this case to avoid any confusion): You didn't rectify this as well. I do not see any changes between the first and new manuscript version.
- Your manuscript still needs read proof. I still see some dot points at the beginning of sentences at the beginning of paragraphs as well as non complete signs (%) in the tables.
You say you are studying only 7 / 8 papers from the abstract, though you reference other papers in the results section. This is not consistent, either you limit your paper to the 7/8 mentioned papers or you increase the number of papers you are referencing all over the sections of your manuscript: I do not see you took care of this comment as well . Other papers you stated are just staked without a clear purpose. Define a clear purpose of including more papers than the seven ones you included in your metadata analysis.
- When you highlight a text in the manuscript for the new version please associate between things you change and my exact comment. For now you are just highlighting some random text that I do not see is associated clearly with my comments.
Round 3
Reviewer 1 Report
Comments and Suggestions for Authors
In the background section, most examples serve as mere expansions of knowledge base.Citing 1-2papers for those examples is enough.
Round 4
Reviewer 1 Report
Comments and Suggestions for Authors
The manuscript has been revised accordingly, and it is now recommended for publication.